# Comparison Study of an Optimized Ultrasound-Based Method versus an Optimized Conventional Method for Agar Extraction, and Protein Co-Extraction, from *Gelidium sesquipedale*

**DOI:** 10.3390/foods11060805

**Published:** 2022-03-11

**Authors:** Laura Pilar Gómez Barrio, Eduarda Melo Cabral, Ming Zhao, Carlos Álvarez García, Ramsankar Senthamaraikannan, Ramesh Babu Padamati, Uma Tiwari, James Francis Curtin, Brijesh Kumar Tiwari

**Affiliations:** 1Department of Food Chemistry and Technology, Teagasc Food Research Centre, Ashtown, D15 DY05 Dublin, Ireland; laurapilar.gomezbarrio@teagasc.ie; 2School of Food Science and Environmental Health, College of Science and Health, Technological University Dublin, D07 ADY7 Dublin, Ireland; uma.tiwari@tudublin.ie (U.T.); james.curtin@tudublin.ie (J.F.C.); 3Department of Food Quality and Sensory Science, Teagasc Food Research Centre, Ashtown, D15 DY05 Dublin, Ireland; eduarda.neves@teagasc.ie (E.M.C.); carlos.alvarez@teagasc.ie (C.Á.G.); 4School of Biosystems and Food Engineering, University College Dublin, Belfields, D04 V1W8 Dublin, Ireland; ming.zhao@ucd.ie; 5School of Chemistry, AMBER Centre, Trinity College Dublin, D02 PN40 Dublin, Ireland; ramsankar.ece@gmail.com (R.S.); babup@tcd.ie (R.B.P.)

**Keywords:** agar, ultrasound, macroalgae, optimization, extraction

## Abstract

Agar is a hydrocolloid found in red seaweeds, which has been of industrial interest over the last century due to its multiple applications in the food, cosmetic, and medical fields. This polysaccharide, extracted by boiling for several hours, is released from the cell wall of red seaweeds. However, the environmental impact coming from the long processing time and the energy required to reach the targeted processing temperature needs to be reduced. In this study, a response surface methodology was employed to optimize both conventional extraction and ultrasound-assisted extractions. Two different models were successfully obtained (R^2^ = 0.8773 and R^2^ = 0.7436, respectively). Additionally, a further re-extraction confirmed that more agar could be extracted. Protein was also successfully co-extracted in the seaweed residues. Optimized conditions were obtained for both the extractions and the re-extraction of the two methods (CE: 6 h, 100 °C; and UAE: 1 h, 100% power). Finally, FT-IR characterization demonstrated that the extracts had a similar spectrum to the commercial agar. Compared to commercial samples, the low gel strength of the agar extracts shows that these extracts might have novel and different potential applications.

## 1. Introduction

Green industries have gained importance over the last decades as an alternative to climate-change-contributing industries, specially the industry extracting hydrocolloids from marine resources, which already has been categorized as sustainable production [1]. Hydrocolloid extraction research has been focussing on novel extraction technologies with the aim of reducing the time of extraction, organic solvents use, and power input. These novel extraction techniques include ultrasound-assisted extraction (UAE) [2], microwave-assisted extraction (MAE) [3], pressurized liquid extraction (PLE) [4], photo-bleaching [5], or reactive extrusion [6]. These novel extraction technologies have been used to extract hydrocolloids such as agar, alginates, fucoidan, and carrageenans. Among these polysaccharides, agar is one of the most-produced hydrocolloids due to the different applications for which it is used (e.g., thickening and gelling agent, growth media, or 3D impression moulds) [7,8], and its economical relevance. From 2009 to 2015, an annual average growth rate of 7% was reported for agar sales, its sale price increasing by 6% within the same time frame [1].

Agar is a polysaccharide formed by β-1,3-linked galactopyranose and α-1,4-linked 3,6-inner ether-L-galactopyranose [9]. To date, this polysaccharide has been extracted at an industrial scale from red macroalgae by following an already stablished conventional method that consists of (1) an optional alkali pre-treatment to eliminate the sulphate groups in order to increase the gel strength—this step required only for some species of red seaweed (i.e., *Gracilaria* spp.) and not for other species (i.e., *Gelidium* spp.), having other effects as lower yields of extraction; (2) an aqueous extraction of agar at temperatures equal or above 85 °C; (3) filtration; and (4) cooling, freezing, and thawing of the agar gel [10,11]. This extraction procedure requires up to four hours of extraction time depending on the seaweed species, while applying boiling temperatures for such long times may compromise the quality of the agar extracts in terms of rheological properties due to polymer degradation [7]. This traditional method has the potential to be improved by using novel extraction technologies, which can lead to higher reproducibility, higher extraction yields, better extract properties, and less energy-intensive conditions by decreasing the extraction times. This can have a final considerable impact on the price of the energy, as shown when comparing the price per gram of the alginate extracted by ultrasound (0.04 €/g) or by conventional extraction (0.20 €/g) [12]. On the other hand, the residues generated are usually overseen and discarded or disposed. In order to increase the sustainability of the overall process and make better use of the raw material, its use as a source of proteins should be considered. Therefore, we aimed to develop a bio-refinery approach where the least possible waste is generated, which is better for the environment.

As previously mentioned, novel extraction technologies aim to transform traditional and non-sustainable methods into greener methods to extract hydrocolloids. In this study, UAE was used to extract agar from *Gelidium sesquipedale*, which is a common source of agar. This technology is based on the cavitation phenomenon that takes place through the formation of cavitation bubbles on the surface of the cell walls. The ultrasound action on the plant materials is reported to degrade the plant matrix into smaller particles with an increased surface contact with the solvent; thus, giving a better interaction between the solvent and compounds of interest. It also increases the size of the plant matrix pores, leading to a more efficient mass transfer compared to conventional methods [13]. Following ultrasonic treatments, the principle of solid–liquid extraction is that the soluble compounds from the plant material dissolve in the solvent of choice when the plant material is in contact with the same. At the same time, the mass transfer of the soluble compounds takes place through a concentration gradient [14].

The main objective of this study is to optimize the traditional agar extraction method, and to compare the results to an alternative, optimized novel extraction method: UAE from *Gelidium sesquipedale*. In order to optimize these processes, a statistical design approach based on Response Surface Methodology (RSM) was used [15]. Optimized conditions were obtained for both experiments, and the extracts obtained were characterized using FT-IR and gel strength. In order to pursue the least-waste approach, the residues generated after agar extraction were characterized according to their amino acid profile and protein content, looking for further applications of the entire seaweed biomass.

## 2. Materials and Methods

### 2.1. Raw Seaweed Material

Dried *G. sesquipedale* was kindly donated by Hispanagar (Burgos, Spain) from a unique batch to ensure the reproducibility of this study. The raw material was washed with tap water until no sand leftovers were found in the washing water. The raw material was then placed in aluminium trays in a 60 °C oven overnight to dry until reaching to constant weight.

### 2.2. Agar Extraction

The agar extraction from *G. sesquipedale* was performed by either the traditional method or following the UAE methodology.

### 2.3. Agar Conventional Extraction Method

A sample of 20 g of *G. sesquipedale* was mixed with 500 mL of distilled water inside a round bottom flask inside an oil bath (LAUDA E100 ecoline 011) and attached to a reflux condenser to prevent water evaporation. Different extraction times and temperatures were performed, as indicated by the RSM design (Table 1). After the extraction was completed, all the samples were treated equally as follows: the mixture was filtered through a muslin cloth while it was still hot, and the filtrate was placed in an aluminium tray to gellify at room temperature; the retentate (residue) was kept for further analysis. Once the gel was formed, the sample was kept in a freezer (UltraLow Temperature Freezer, Haier Medical and Laboratory Products Co., Ltd., Qingdao, China) (−20 °C) overnight. In the following day, the samples were defrosted, and the gel was squeezed manually to eliminate most of the water. The squeezed gel was then kept in a freezer (−80 °C), and then freeze-dried (−25 °C/30 °C, approximately 0 mbar) in an FD 80 model (Cuddon Engineering, Blenheim, New Zealand) to obtain the final dried agar extracts, which were used for further analysis.

Afterwards, a re-extraction was performed on the residues from the previous extraction by soaking 10 g of the dried residue in 200 mL in distilled water during 2 h at 95 °C inside a water bath. After agar extraction, the mixture was processed as per the previously described processing procedures. The performed methodology is illustrated in Figure 1.

### 2.4. Agar Ultrasound-Assisted Extraction

A sample of 20 g of *G. sesquipedale* was mixed with 500 mL of distilled water inside a beaker placed inside a water bath where the temperature was kept at 85 °C. The probe transducer (20 kHz, 1000 UHdt Hielscher Ultrasound Technology, Teltow, Germany) was placed inside the mixture. Different extraction times and ultrasound probe powers were performed as indicated by the RSM design (Table 2). After the ultrasound treatment, the samples were treated as previously described in Section 2.3. The performed methodology is illustrated in Figure 2.

### 2.5. Determination of Crude Agar Yield of Extraction and Re-Extraction

The freeze-dried agar samples were weighted down, and the crude agar yield was calculated using the formula in Equation (1) (WFDA: weight freeze-dried agar, and WIRS: weight initial raw seaweed) and Equation (2). (WRAE: weight residue after extraction):(1)$$\mathrm{Yield}\;\mathrm{crude}\;\mathrm{agar}\;\mathrm{extraction}\;\%\;\left(\frac{w}{w}\right)=\frac{\mathrm{WFDA}\;\left(g\right)}{\mathrm{WIRS}\;\left(g\right)}\times 100$$
(2)$$\mathrm{Yield}\;\mathrm{of}\;\mathrm{reextraction}\;\%\;\left(\frac{w}{w}\right)=\frac{\mathrm{WFDA}\;\left(g\right)}{\mathrm{WRAE}\;\left(g\right)}\times 100$$

### 2.6. Experimental Design and Statistical Analysis of Conventional Agar Extraction, and Ultrasound-Assisted Extraction

A two level, three-variable Composite Central Design (CCD) was employed in this study, requiring a total of 13 experiments for the optimization of traditional extraction parameters (5 central points). The parameters and levels employed were the extraction time (2–6 h) and temperature (80–110 °C). The coded and original values of the independent variables used in this experiment are listed in Table 1.

A two level, three-variable Composite Central Design (CCD) was employed in this study, requiring a total of 13 experiments for the optimization of the UAE parameters (3 central points). The parameters and levels employed were extraction time (30–60 min), and power (50–100%). The coded and original values of the independent variables used in this experiment are listed in Table 2. 

The extraction yield of the agar from the CCD, together with three other responses (yield of re-extraction (%), protein extraction (%), and protein re-extraction (%)), were analysed using the response surface regression (Table 3 and Table 4) and fitted to a second-order polynomial model (Equation (3)). Minitab 17 was the software used for Response Surface Methodology. An ANOVA test and an F test were performed to exam the significance of the experimental data. The validity of the model was determined by comparing the experimental and predicted values.
(3)$$Y=\beta_{0}+\sum_{j=1}^{k}\beta_{j}x_{j}+\sum_{j=1}^{k}\beta_{\mathrm{jj}}x_{j}^{2}+\sum \sum_{j<j=2}^{k}\beta_{\mathrm{ij}}x_{i}x_{j}$$
where the responses, x_i_ and x_j_, are the independent variables, β_0_ stands for the constant term, β_j_ stands for the linear coefficient, β_ij_ stands for the interaction coefficient, β_jj_ stands for the quadratic factor, and k is the number of independent parameters.

### 2.7. Protein Analysis and Amino Acid Profile

The analysis was performed as explained by Hildebrand et al. [16] using the same system described in this work. Briefly, the analysis was performed using an UHPLC-FLD instrument (Thermo Ultimate 3000 RS, Thermo Scientific, Waltham, MA, USA) equipped with an Agilent AdvanceBio AAA column (100 mm length × 3.0 mm internal diameter × 2.7 μm particle size, Agilent Technologies, Santa Clara, CA, USA). The A mobile phase consisted of 10 mM Na_2_HPO_4_ in 10 mM Na_2_B_4_O_7_ · H_2_O (pH 8.2), and the B mobile phase consisted of a mixture of acetonitrile, methanol, and water (45:45:10, *v*:*v*:*v*).The protein was hydrolysed to determine the total amino acid content. These analyses were conducted on the residues generated after the first extraction and the following re-extraction. The protein was calculated based on the total amino acid content.

### 2.8. Gel Strength

Gel strength measurements of the agar extracts obtained from the optimized conditions for conventional agar extraction and the ultrasound-assisted extraction were based on penetration tests on the formed agar gels using gel fracture force to determine the gel strength. For this purpose, 1.5% (*w*/*w*) solutions of agar in distilled water were prepared at 95 °C until all the soluble compounds were dissolved, and then transferred to a bloom jar and kept in a fridge at 4 °C overnight. The next day, the samples were left to reach room temperature before the analysis. A texture analyser (Stable Micro Systems model—TA.HD plus C, Surrey, UK) equipped with a radiused cylinder probe (P/0.5 R, 1.27 cm diameter) was operated at a penetration rate of 1 mm/s to a depth of 5 mm on the formed gel as performed by [17]. Each sample was measured in duplicates.

### 2.9. Fourier Transform Infrared Spectroscopy (FT-IR)

The agar extracts were analysed using a PerkinElmer Frontier FT-IR spectrometer. Single beam reflectance spectra (R) were collected over the wavenumber range 600–4000 cm^−1^ with a resolution of 4 cm^−1^, and then were converted and recorded as transmittance.

## 3. Results and Discussion

### 3.1. Modelling of the Extraction Process

#### 3.1.1. Conventional Extraction Method

The four response dependent variables (yield of extraction (%), yield of re-extraction (%), protein extraction (%), and protein re-extraction (%)) for each run in the experimental design are listed in Table 1. The yield of extraction ranged from 1.50% to 15.85%; yield of re-extraction from 13.47% to 23.14%; the protein extraction from 9.22% to 12.42%; and the protein re-extraction from 12.09% to 26.14%. A linear model fitted well the experimental data only in the case of the response of the yield of extraction with a low standard error (*p* < 0.001) and a coefficient of regression of 0.8773. In the case of the other responses studied, it can be seen on the 3D figure that none of the parameters studied affects the yield of protein re-extraction. In the case of the protein extraction response, it was found that only temperature was significant, while a longer time shows a trend in where it may have a positive effect. This may be explained based on the fact that even the mildest time and temperature conditions here tested were sufficient to reach the highest protein yield extraction. Regression analysis was performed on the experimental data and the coefficients of the model were evaluated for statistical significance using ANOVA analysis.

Equation (4) describes the relationship between the extraction time and temperature for the yield of the extraction, the yield of the re-extraction, the protein extraction, and the protein re-extraction. 3D surface plots illustrate the relationship between the yield of the extraction (Figure 3A), the yield of the re-extraction (Figure 3B), the protein extraction (Figure 3C), the protein re-extraction (Figure 3D), and the experimental variables—the time and temperature of the extraction.
Yield = −35.80 + 2.207 Time (h) + 0.3482 Temperature (°C)(4)

As the quadratic terms were not significant, the model for the “yield of extraction (%)” response was changed into a linear model (R^2^ = 0.801) with Equation (4). In the case of protein extraction, only temperature was found to be significant and the model regression coefficient was R^2^ = 0.618. The other responses had no significant linear terms. In Figure 3, graphical representations of each of the models are illustrated.

##### Optimized Conventional Agar Extraction and Re-Extraction

The optimized conditions for the conventional agar extraction and re-extraction were obtained based on the highest agar yield achievable. For that purpose, Equation (4) was used to calculate the estimated agar yield of the extraction that could be obtained, also to compare this yield to the one obtained after performing the extraction with the optimal conditions (X_1_: 6 h and X_2_: 110 °C) (Table 5). The estimated yield corresponds to the yields obtained from the validation extraction. Other studies obtained the maximum yield of extraction (43.3%) at 3 h and 100 °C; nevertheless, the ratio used was considerably higher (10 g of *H. cornea* in 1.8 L) compared to this study (20 g of *G. sesquipedale* in 500 mL) and the differences in the yields could be observed due to this [18].

#### 3.1.2. Ultrasound-Assisted Extraction (UAE)

The four response dependent variables (yield of extraction (%), yield of re-extraction (%), protein extraction (%), and protein re-extraction (%)) for each run in the experimental design are listed in Table 2. The yield of extraction ranged from 4.39% to 6.82%; yield of re-extraction from 8.50% to 13.85%; the protein extraction from 10.12% to 12.02%; and the protein re-extraction from 11.16% to 14.56%. A linear model fitted the experimental data only in the case of the response of the yield of extraction, with low standard error (*p* < 0.01) and a coefficient of regression of 0.7436. It can be seen in Figure 4A that lower power led to lower yields, the same as what lower time led to lower yields. The models developed for the other responses did not show any significance, which could be explained because the parameters of the model were not chosen to optimize these responses (re-extraction, and protein extraction and re-extraction), and also because of the possible depolymerisation caused by ultrasound treatment [7]. The significance of the two experimental variables (time and power) that affect the extraction and re-extraction yields were determined by using Design Expert software. Regression analysis was performed on the experimental data and the coefficients of the model were evaluated for statistical significance using ANOVA analysis. 3D surface plots illustrate the relationship between the yield of extraction (Figure 4A), the yield of re-extraction (Figure 4B), the protein extraction (Figure 4C), the protein re-extraction (Figure 4D), and the experimental variables—the time of extraction and the ultrasound power. In Figure 4, graphical representations of each one of the models is illustrated.

As the quadratic terms were not significant, the model for the “yield of extraction (%)” response was changed into a linear model (R^2^ = 0.5338) with the following Equation (5). The other responses had no significant linear terms.
Yield (%) = 2.598 + 0.0382 Time (min) + 0.01388 Power (%)(5)

##### Optimized Ultrasound-Assisted Extraction and Re-Extraction

The optimized conditions for the Ultrasound–Assisted Extraction and re-extraction were obtained based on the highest agar yield achieved. For that purpose, Equation (5) was used to calculate the estimated agar yield of the extraction that could be obtained, also to compare this yield to the one obtained after performing the extraction with the optimal conditions (X_1_: 60 min and X_2_: 100% ultrasound power) (Table 6). From Equation (5), it was calculated that the agar yield of the extraction under optimum conditions would be 6.70%. The yield obtained following the optimum conditions for UAE represent an additional two percent compared to the estimated yield for the same conditions. No studies about the optimization of the agar extraction by means of ultrasound were found; nevertheless, Martínez-Sanz et al. [19] extracted agar from *G. sesquipedale* by applying Ultrasound-Assisted Extraction for 1 h (100% power, 25 kHz), reporting a yield of extraction of 18.5 ± 2.4% using a 1:25 ratio, higher than the yield obtained for this experiment, where the only difference is the frequency of the ultrasound probe (20 kHz). This finding shows that a higher frequency could have a potential impact on the yields of extraction.

### 3.2. Effect of Independent Variables on the Responses of the Processes

#### 3.2.1. Conventional Extraction

Extraction time only significantly affected the yield of extraction (*p* < 0.01), whereas extraction temperature significantly affected the yield of extraction (*p* < 0.001) and the protein extraction (*p* < 0.05). As a result, linear effects of the independent variables were dependent for the protein of extraction, and only temperature in the case of the yield of re-extraction. However, in the case of yield of re-extraction and protein re-extraction, analysis regression of the model shows that the linear, quadratic, and interaction regression did not fit the experimental data well. Nevertheless, it is seen that the re-extraction achieved extracting more agar from that already used for extraction of the *G. sesquipedale* sample, and that the protein was also co-extracted in the residues from both the extraction and re-extraction. No relation was found among the samples with the lowest extraction yields, and the samples with the highest re-extraction yields. The highest yield of extraction (15.85%) was obtained at 5 h and 105 °C; the highest yield of re-extraction (21.85%) was obtained for the re-extraction of the 4 h and 95 °C samples; the highest protein content for extraction (11.80%) was obtained for the sample treated at 3 h and 105 °C; and the highest protein content for re-extraction (26.14%) was obtained for the sample treated at 3 h and 85 °C. At higher temperatures and times of extraction, higher yields of extraction were achieved; the same trend can be seen for the protein content of extraction whereas the yield of re-extraction was not significantly affected by the increase in temperature, and the protein content of re-extraction was not affected by any of the experiment variables (Figure 3A–D). As seen in the literature, the conditions required for higher yields of agar extraction from *Gelidium* spp. are temperatures of extraction of 105–110 °C for 2–4 h, which supports the evidence found in this study [11].

#### 3.2.2. Ultrasound-Assisted Extraction

Extraction time (*p* < 0.01) affected significantly only the yield of extraction (%), whereas extraction power did not affect significantly the yield of extraction (%) (*p* > 0.05). These independent variables did not significantly affect any of the other responses, which are yield of re-extraction (%), protein extraction (%), and protein re-extraction (%). In the case of yield of extraction (%), protein extraction (%), and protein re-extraction (%), analysis of the regression model show that the linear, quadratic, and cross product (interaction) did not fit the experimental data well. The same findings as for the conventional extraction for the yields of re-extraction and for the protein yields were found for the UAE. The highest yield of extraction (6.82%) was obtained at 60 min and 75% ultrasound power; the highest yield of re-extraction (16.14%) was obtained for the re-extraction of the 60 min and 50% ultrasound power samples; the highest protein content for extraction (12.02%) was obtained for the sample treated at 30 min and 100% ultrasound power; and the highest protein content for re-extraction (14.56%) was obtained for the sample treated at 45 min and 50%. For all the responses, a higher time of extraction contributed to a higher yield, except for the yield of protein extraction (%) as at short times all the extractable protein was already solubilized. In turn, higher ultrasound power did not seem to have a significant effect on any of the responses (Figure 4A–D). A similar finding was reported on another study where the density of power (W cL^−1^) did not have an effect on the yield of the polysaccharide extracted from *Silvetia compressa* [20]. No studies were found where the effect of UAE treatment was studied for agar extraction; nevertheless, the UAE time significantly affected the yield of the alginate extraction from *Sargassum muticum*, increasing from 5.7 to 15% from 5 min to 30 min of extraction [12].

### 3.3. Fourier Transform–Infrared Analysis

The FT-IR analysis was performed in order to confirm that the chemical bonds normally found in agar were present in the extracts. It was also used to identify possible impurities found in the same. The FT-IR spectra of the agar extracts and commercial agar (Sigma agar product (CAS-No: 9002-18-0), Sigma-Aldrich, Wicklow, Ireland were collected and ranged from 600 to 4000 cm^−1^, presented in Figure 5. As mentioned previously, agar is a polysaccharide formed by β-1,3-linked galactopyranose and α-1,4-linked 3,6-inner ether-L-galactopyranose [9]. Two specific bonds to these two compounds can be identified in the spectral features at 891 cm^−1^ and 930 cm^−1^, which have been related to the C–H bond of the β-1,3-linked galactopyranose and the 3,6-anhydro-galactose residue, respectively. As reported for all polysaccharides, a strong peak at 1041 cm^−1^ and a medium peak at 1151 cm^−1^ are shown in the spectra, which corresponds to the C–O and C–C stretching vibrations of the pyranose ring [21]. A strong peak at 1646 cm^−1^ is reported to be related to the N–H impurity that belongs to the peptidic bond in proteins [22]. Finally, between 3000 cm^−1^ and 3600 cm^−1^, a broad peak reported for the O–H stretching can be found for all the extracts and agar commercial sample [23].

The spectral shape of all the agar extracts shown in Figure 5 and Figure 6 is similar to the commercial agar sample.

### 3.4. Gel Strength

The different agar extracts obtained following the optimized conditions for conventional extraction (6 h, 110 °C) and ultrasound assisted extraction (60 min, 100% power) were tested for gel strength following a penetration test. Table 7 shows the gel strength values obtained for the agar extracts from different optimized methods. The highest gel strength was reported for the commercial agar (523 ± 36.79 g/cm^2^) while the lowest gel strength was reported for Ultrasound-Assisted Extraction (26.84 ± 15.6 g/cm^2^). This finding is opposite to the initially expected results of a higher gel strength for the Ultrasound-Assisted Extraction samples compared to the conventional extraction samples. No negative impact from the six hours of extraction of the conventional method was found. A 750 g/cm^2^ value for gel strength is considered a high-quality agar; therefore, none of the agar extracts can be considered a high-quality agar [24]. Furthermore, a higher gel strength is directly correlated with a higher purity of the agar extracts and a lower content of sulphate groups; therefore, it seems that, in general, the sulphate group concentration is high in all samples but especially so in the ultrasound-treated samples [25]. This higher sulphate content could be explained due to the presence of other compounds that may have not been co-extracted following the conventional extraction. This effect was observed when other bioactives were extracted [26]. Nevertheless, molecular weight is also related to the gel strength as a higher molecular weight leads to a higher gel strength. A depolimerisation process may have occurred and led to the low gel strength of the ultrasound-treated samples [27]. The re-extraction extracts seemed to have statistically similar (*p* < 0.05) gel strength compared to the extraction extracts, being in the case of the Ultrasound-Assisted re-extraction higher than the Ultrasound-Assisted Extraction, which could suppose that additional agar was extracted. A study where agar was also extracted by conventional extraction and by Ultrasound-Assisted Extraction reported the following values, 377 ± 136 g/cm^2^ and 438 ± 47 g/cm^2^, respectively; compared to this study, a similar gel strength for the conventional extraction was obtained (394.93 ± 81.93 g/cm^2^), whereas a much lower gel strength was obtained for the Ultrasound-Assisted Extraction (26.84 ± 15.6 g/cm^2^) [19]. 

### 3.5. Amino Acid Profile of Gelidium Sesquipedale Extraction Residues

The residues obtained after performing conventional agar extraction and Ultrasound-Assisted Extraction were characterised to obtain their essential amino acid profiles. In Table 8 and Table 9, the total amino acid concentration (mg/g) in the residues is shown for the conventional agar extraction and ultrasound-assisted extraction, respectively. The re-extraction protocol concluded on higher concentrations of total amino acids compared to extraction protocols for both conventional and Ultrasound-Assisted Extraction. This finding showed that the residues obtained from the agar extraction protocols had recoverable protein, which had not been extracted, and by means of a re-extraction process could be extracted. Nevertheless, no significant relationship (*p* > 0.05) between the extraction time or temperature applied, and the total concentration of amino acids was found, as observed also for total protein yields, which supports these findings. If the re-extraction samples are compared, it can be seen that the highest total amino acid concentration was obtained for the (3 h, 85 °C) sample. In the case of UAE, it was found that not the power nor the time had any significant effect (*p* > 0.05) on the total concentration of amino acids. In all cases, the highest amino acid concentration was found for aspartic acid, which agrees with what was found by Trigueros et al. [28] after using subcritical water extraction from *G. sesquipedale*.

## 4. Conclusions

Two Response Surface Methodology studies were conducted to optimise the agar extraction processes by means of conventional or Ultrasound-Assisted Extraction; also, we investigated the residues generated as a source of proteins to be re-extracted, aiming to obtain other valuable compounds, such as amino acids and proteins. In this study, models were successfully obtained for both conventional extraction and UAE, with a regression coefficient of 0.8773 and 0.7436, respectively. Optimized conditions were also successfully obtained (CE: 6 h, 100 °C, and UAE: 1 h, 100% power). It has been found that increasing time and temperature for conventional extraction, or increasing time for Ultrasound-Assisted Extraction, can result in higher yields in agar extraction; nevertheless, ultrasound power did not affect the agar yields of extraction. Especially for the Ultrasound-Assisted Extraction method, the seaweed cell walls can be effectively damaged, and therefore lead to a more efficient mass transfer. On the other hand, protein extraction was not affected by the conditions here studied. Additional re-extraction methodologies for agar and protein were analysed, too, although the models developed were not significant. The reason behind a better fit of the conventional extraction model could be that the parameters tested were synergetic, whereas the effect of ultrasound power is not so clear, as it does not show a clear trend on the yields. This could be explained as the parameters were not chosen based on enhancing these responses (re-extractions and protein extractions). The characterization of the optimized extracts confirmed that agar was extracted as the spectra obtained for the FT-IR matched the spectrum of the commercial agar. The gel strength of the conventional extracts was significantly higher than US extracted ones, although not as high as the commercial sample. Nevertheless, this low gel strength could be used in novel applications such as dressings, or fat replacers. Further studies focusing on testing ultrasound action on the cell walls are required to figure out the conditions to minimize agar depolymerisation. Re-extraction protein protocols led to higher concentrations of total essential amino acids compared to extraction protein protocols, for conventional and ultrasound-assisted methods. Based on our study, it can be concluded that UAE would allow to extract a higher yield compared to conventional extraction when the same extraction time is used; therefore, it is more convenient from an economical point of view, as less energy is required (Appendix A, obtained by using a power meter set with a pre-defined value for the cost of the electricity of 0.1 €/kW·h). It would also be of interest to have more studies focussed on co-extraction of additional, industrially valuable extracts, to make better use of the sources used for agar extraction.

## Figures and Tables

**Figure 1 foods-11-00805-f001:**
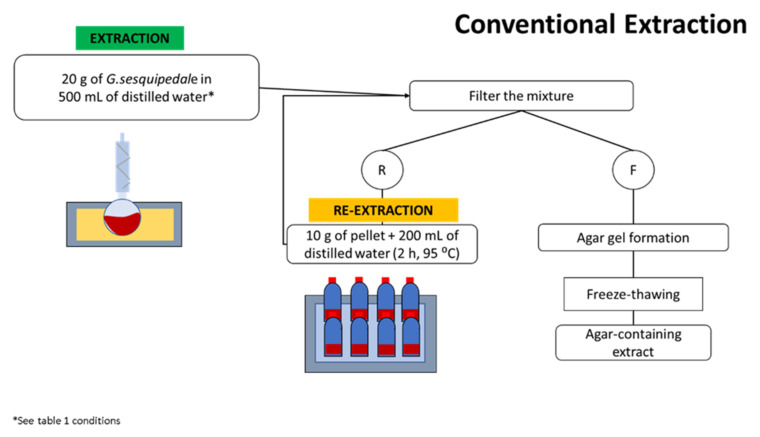
Workflow of the conventional extraction method performed for agar extraction (R: seaweed residue; and F: filtrate).

**Figure 2 foods-11-00805-f002:**
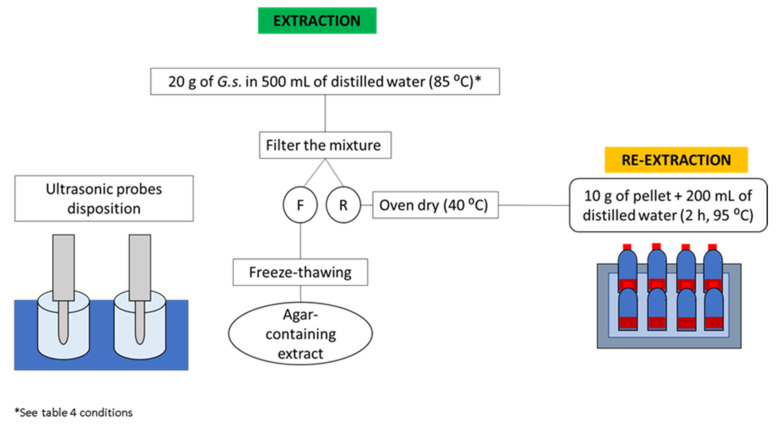
Workflow of the ultrasound-assisted extraction performed for agar extraction (F: filtrate, and R: seaweed residue).

**Figure 3 foods-11-00805-f003:**
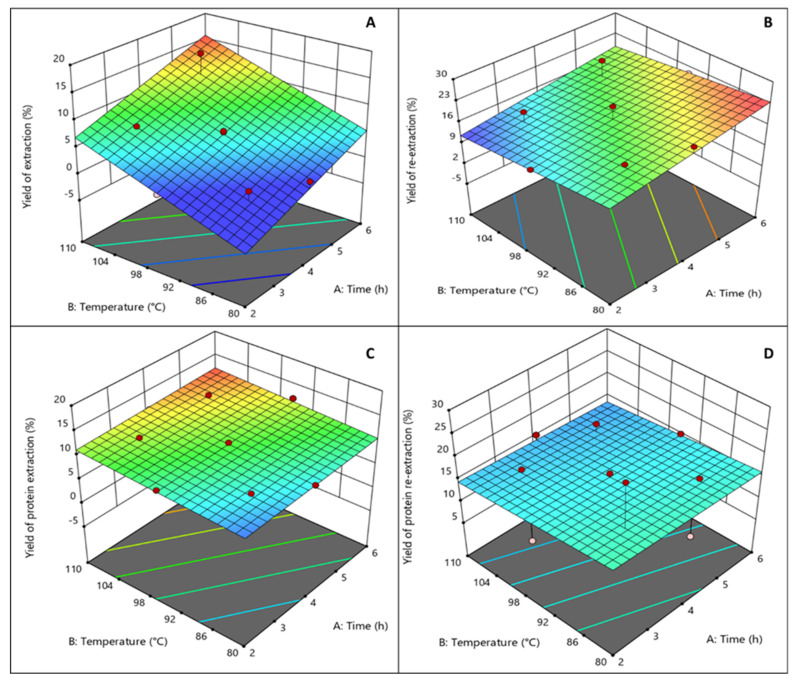
Design Expert 3D conventional agar extraction surface graphs ((**A**) yield extraction (%), (**B**) yield re-extraction, (**C**) protein extraction (%), and (**D**) protein re-extraction (%)).

**Figure 4 foods-11-00805-f004:**
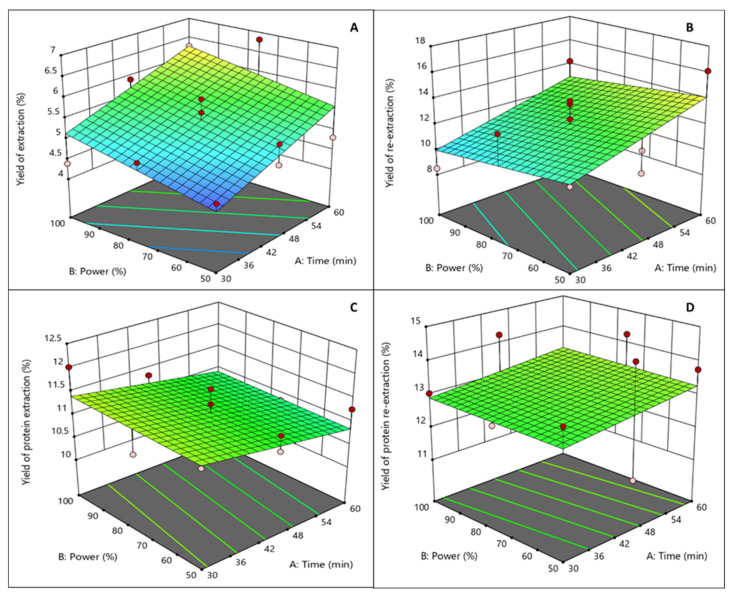
Design Expert 3D ultrasound-assisted agar extraction surface graphs ((**A**) yield extraction (%), (**B**) yield re-extraction, (**C**) protein extraction (%), and (**D**) protein re-extraction (%)).

**Figure 5 foods-11-00805-f005:**
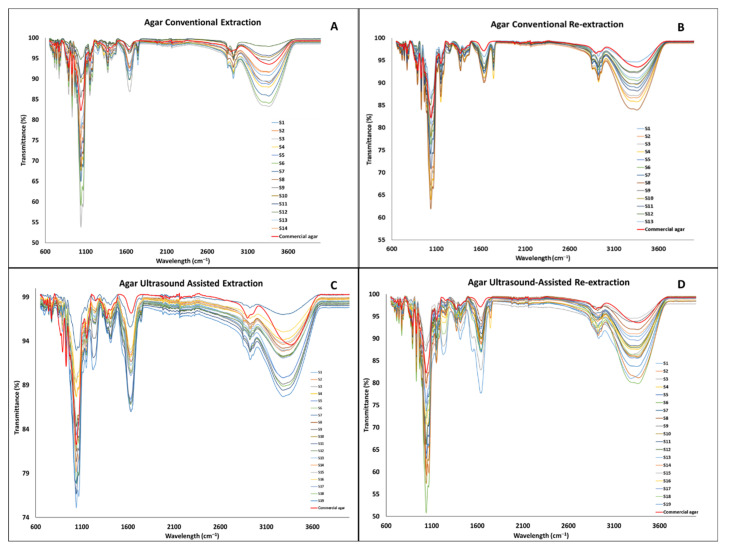
FT-IR spectra of the agar extracted by different methods ((**A**) agar conventional extraction, (**B**) agar conventional re-extraction, (**C**) agar ultrasound-assisted extraction, (**D**) agar ultrasound-assisted re-extraction).

**Figure 6 foods-11-00805-f006:**
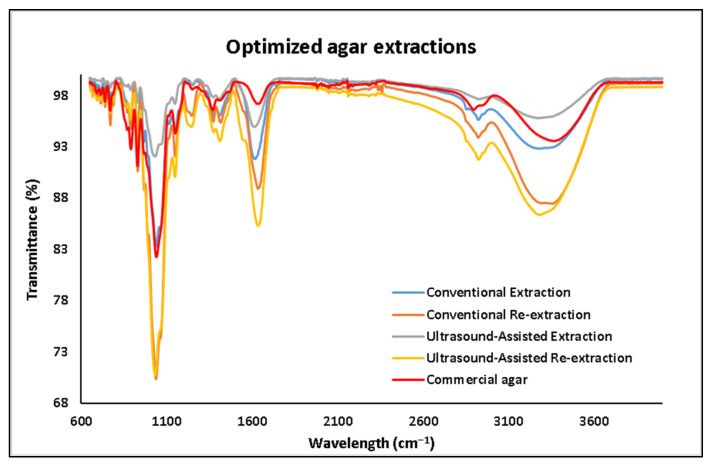
FT-IR spectra of the agar extracted by different optimized methods.

**Table 1 foods-11-00805-t001:** Response surface central composite design (coded and uncoded) and the results of the conventional extraction and re-extraction yield of agar.

No.	Extraction Time (h)	Extraction Temperature (°C)	Yield of Extraction (%)	Yield of Re-Extraction (%)	Protein Extraction (%)	Protein Re-Extraction (%)
1	2	95	1.5	14.31	10.71	11.86
2	3	85	2.15	18.53	10.52	26.14
3	5	85	3.45	20.98	10.44	16.46
4	3	105	7.53	17.79	11.80	15.55
5	5	105	15.85	23.14	12.42	16.08
6	4	95	6.69	15.43	9.22	13.21
7	4	95	5.19	21.85	11.21	16.12
8	6	95	9.93	21.09	12.10	15.13
9	4	80	2.2	22.1	10.06	12.43
10	4	95	6.49	14.18	10.51	12.09
11	4	95	5.98	13.47	10.76	14.41
12	4	110	10.08	12.16	11.25	15.27
13	4	95	2.4	16.07	10.76	14.36

**Table 2 foods-11-00805-t002:** Response surface central composite design (coded and uncoded) and the results for the ultrasound-assisted extraction and re-extraction yield of agar.

No.	Extraction Time (min)	Power (%)	Agar Yield of Extraction (%)	Agar Yield of Re-Extraction (%)	Protein Extraction (%)	Protein Re-Extraction (%)
1	45	50	5.29	10.04	11.3	14.56
2	45	100	5.91	9.84	11.34	14.25
3	60	75	5.44	11.46	10.85	14.27
4	30	75	4.97	12.97	10.83	12.76
5	60	50	4.81	16.14	11.27	13.75
6	30	50	4.62	11.03	11.27	13.45
7	30	100	4.39	8.50	12.02	13.04
8	60	100	6.28	14.23	10.59	12.73
9	45	75	5.18	13.85	10.12	11.83
10	45	75	5.87	12.43	11.63	12.96
11	45	75	5.54	13.6	11.31	12.46
12	45	50	4.79	11.74	10.97	11.16
13	60	75	6.82	12.98	10.85	13.05

**Table 3 foods-11-00805-t003:** Regression coefficients of the predicted second-order polynomial models for the investigated responses from agar extracted by conventional extraction.

Coefficient	Yield of Extraction	Yield of Re-Extraction	Protein Extraction	Protein Re-Extraction
Intercept				
β_0_	88.0 ***	148 ***	29.1 ***	190 ***
Linear				
β_1_	−15.66 **	−9.1 ^ns^	−3.28 ^ns^	−24.9 ^ns^
β_2_	−1.52 ***	−2.30 ^ns^	−0.320 *	−2.56 ^ns^
Quadratic				
β_11_	0.149 ^ns^	0.5 ^ns^	0.002 ^ns^	0.054 ^ns^
β_22_	0.006 ^ns^	0.010 ^ns^	0.018 ^ns^	0.255 ^ns^
Cross product				
β_12_	0.176 ^ns^	0.073 ^ns^	0.248 ^ns^	-
R^2^	0.877	0.426	0.618	0.234
*p* value	0.002 **	0.463 ^ns^	0.159 ^ns^	0.816 ^ns^

* Significant at *p* ≤ 0.05; ** Significant at *p* ≤ 0.01; *** Significant at *p* ≤ 0.001; ns = not significant.

**Table 4 foods-11-00805-t004:** Regression coefficients of the predicted second-order polynomial models for the investigated responses from agar extracted by Ultrasound-Assisted Extraction.

Coefficient	Yield of Extraction (%)	Yield of Re-Extraction (%)	Protein Extraction (%)	Protein Re-Extraction (%)
Intercept				
β_0_	1.90 ***	6.6 ***	10.72 ***	17.43 ***
Linear				
β_1_	0.038 *	−0.169 ^ns^	0.063 ^ns^	−0.098 ^ns^
β_2_	0.0409 ^ns^	0.230 ^ns^	−0.0247 ^ns^	−0.078 ^ns^
Quadratic				
β_11_	−0.00095 ^ns^	0.00253 ^ns^	−0.00007 ^ns^	0.00159 ^ns^
β_22_	−0.000520 ^ns^	−0.00186 ^ns^	0.000466 ^ns^	0.00065 ^ns^
Cross product				
β_12_	0.001130 ^ns^	0.00041 ^ns^	−0.000953 ^ns^	−0.00040 ^ns^
R^2^	0.7436	0.4327	0.4309	0.1232
*p* value	0.047 *	0.451 ^ns^	0.454 ^ns^	0.954 ^ns^

* Significant at *p* ≤ 0.05, *** Significant at *p* ≤ 0.001, ns Not significant.

**Table 5 foods-11-00805-t005:** Agar yield obtained from the optimized conventional extraction (*n* = 2).

Extraction Method	Agar Yield of Extraction (%)	Agar Estimated Yield of Extraction (%)
Conventional extraction	16.55 ± 0.73	15.74

**Table 6 foods-11-00805-t006:** Agar yields obtained from optimized Ultrasound-Assisted Extraction and re-extraction (*n* = 2).

Extraction Method	Agar Yield of Extraction (%)	Agar Estimated Yield of Extraction (%)
Ultrasound-Assisted Extraction	8.55 ± 0.16	6.70

**Table 7 foods-11-00805-t007:** Gel strength values of the agar extracted by different optimized methods.

Extraction Method	Gel Strength (g/cm^2^) ± SD
Agar Conventional Extraction	394.93 ± 103.0 ^a,b^
Agar Conventional Re-extraction	217.23 ± 48.1 ^b,c^
Agar Ultrasound-Assisted Extraction	26.84 ± 19.8 ^c^
Agar Ultrasound-Assisted Re-extraction	115.26 ± 0.112 ^c^
Commercial Agar	523.24 ± 52.0 ^a^

One-way ANOVA: All the means that do not share a superscript letter are statistically different (*p* ≤ 0.05), calculated by Tukey tests.

**Table 8 foods-11-00805-t008:** Total essential amino acid profile for the conventional extraction and re-extraction.

No.	Extraction Time (h)	Extraction Temperature (°C)	Total AA (mg/g) Extraction ± SD	Total AA (mg/g) Re-Extraction ± SD
1	2	95	124.20 ± 0.17 ^a,b,c,d^	137.55 ± 0.75 ^b^
2	3	85	122.06 ± 3.30 ^b,c,d^	303.22 ± 22.46 ^a^
3	5	85	121.10 ± 1.30 ^b,c,d^	190.91 ± 0.64 ^b^
4	3	105	136.83 ± 1.31 ^a^	180.36 ± 12.53 ^b^
5	5	105	144.03 ± 6.87 ^a^	186.55 ± 1.21 ^b^
6	4	95	106.92 ± 2.93 ^d^	153.24 ± 0.42 ^b^
7	4	95	130.01 ± 6.48 ^a,b,c^	187.01 ± 24.41 ^b^
8	6	95	140.40 ± 3.29 ^a,b^	175.52 ± 0.50 ^b^
9	4	80	116.67 ± 5.28 ^c,d^	144.14 ± 2.22 ^b^
10	4	95	121.88 ± 1.46 ^b,c,d^	140.21 ± 0.31 ^b^
11	4	95	124.87 ± 3.01 ^a,b,c,d^	167.18 ± 0.66 ^b^
12	4	110	130.53 ± 0.47 ^a,b,c^	177.08 ± 12.85 ^b^
13	4	95	-	166.55 ± 4.38 ^b^

One-way ANOVA: All the means that do not share a superscript letter are statistically different (*p* < 0.05), calculated by Tukey tests. This test was performed individually for the extraction samples and for the re-extraction samples.

**Table 9 foods-11-00805-t009:** Total essential amino acid profile for the Ultrasound-Assisted Extraction and re-extraction.

No.	Extraction Time (min)	Power (%)	Total AA (mg/g) Extraction ± SD	Total AA (mg/g) Re-Extraction ± SD
1	60	75	125.89 ± 2.58 ^a^	165.56 ± 6.26 ^a^
2	30	75	122.64 ± 2.98 ^a^	144.80 ± 0.17 ^a^
3	60	50	132.27 ± 8.82 ^a^	152.35 ± 5.79 ^a^
4	30	50	127.70 ± 2.67 ^a^	155.47 ± 4.41 ^a,b^
5	30	100	131.36 ± 2.13 ^a^	147.02 ± 0.41 ^a,b^
6	60	100	122.16 ± 9.00 ^a^	155.39 ± 0.69 ^a,b^
7	45	75	117.38 ± 2.91 ^a^	137.21 ± 5.11 ^a,b^
8	45	75	134.90 ± 9.30 ^a^	144.59 ± 4.85 ^a,b^
9	45	75	131.05± 1.02 ^a^	168.86 ± 4.46 ^a,b^
10	45	50	127.3 ± 4.71 ^a^	129.44 ± 5.47 ^a,b^
11	60	75	125.83 ± 0.56 ^a^	151.37 ± 10.42 ^a,b^
12	45	100	131.52 ± 1.23 ^a^	165.31± 0.39 ^b^
13	30	75	128.65 ± 12.45 ^a^	151.17 ± 1.53 ^a,b^
14	30	50	127.70 ± 2.67 ^a^	155.47 ± 4.41 ^a,b^
15	30	100	147.53 ± 8.00 ^a^	155.42 ± 3.13 ^a,b^
16	60	50	132.27 ± 8.12 ^a^	152.35 ± 5.79 ^a,b^
17	60	100	122.16 ± 8.98 ^a^	155.39 ± 0.69 ^a,b^

One-way ANOVA: All the means that do not share a superscript letter are statistically different (*p* < 0.05), calculated by Tukey tests. This test was performed individually for the extraction samples and for the re-extraction samples.

## Data Availability

Not applicable.

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
