# Peer review of "Comparison Study of an Optimized Ultrasound-Based Method versus an Optimized Conventional Method for Agar Extraction, and Protein Co-Extraction, from Gelidium sesquipedale"

_foods, 2022, doi:10.3390/foods11060805_

Round 1
Reviewer 1 Report
In presented manuscript authors performed the study of two different methods of agar extraction for agar extraction and protein co-extraction, from Gelidium sesquipedale. The topic is interesting and important. However, the article have many drawbacks.
The article is poorly formatted according to Foods standards.
Line 26: Give „2” in R2 in the superscript.
Line 29 Use “°C” not zero in the superscript.
Line 96: Be consequent and write “G. sesquipedale” in Italic form.
Line 82, 97, and in the whole manuscript: use only UAE. This abbreviation was previously defined.
Line 108-109: Please include the conditions of freeze-drying (pressure and temperature)
Each figure should be mentioned in the text. For example, Figures 1 and 2 have no mentioned in the text.
Line 134-136: Please use the symbol in the equations, not text
In the Material and Method, section authors should include information about the statistical evaluation of data.
Why only Anova was performed for values in Tables 7 and 9. What about the data in Tables 1 and 4?
Why the units of strength are expressed in g/cm2. The SI unit should be used as Pascal (N/m2)
Line 182-185 and in the whole manuscript: Do not give the unit (%), just give the unit directly after the numbers.
Lines 505-507 in Conclusions: It is obvious without this study. Compare these two methods of extraction and give a suggestion of which method is better, and suggest the optimized condition of extraction as is written in the title of this paper.
Author Response
The authors appreciate the comments made by Reviewer 1, and wish to acknowledge each of them in the following points. On bold letters the corrections from the reviewer can be found, and following each statement the answer to those corrections. All the corrections have been included using the track mode.
Reviewer 1
In presented manuscript authors performed the study of two different methods of agar extraction for agar extraction and protein co-extraction, from Gelidium sesquipedale. The topic is interesting and important. However, the article have many drawbacks.
The article is poorly formatted according to Foods standards.
The authors are pleased to read that the topic is considered interesting and important, and agree that changes need to be made in order to address those drawbacks. Furthermore, changes in the format have been made in order to present the manuscript according to Foods standards. A more detailed description is shown in the following lines.
Line 26: Give „2” in R2 in the superscript.
The authors have acknowledged this mistake, and it can be found corrected on line 26 now.
Line 29 Use “°C” not zero in the superscript.
The authors have acknowledged this mistake, and it can be found corrected on line 29 now.
Line 96: Be consequent and write “G. sesquipedale” in Italic form.
The authors have acknowledged this mistake, and corrected it. It can be found corrected on lines 70, 82, 90, 96, 99, 123, 285, 357, 374, 484, and 500.
Line 82, 97, and in the whole manuscript: use only UAE. This abbreviation was previously defined.
The authors have acknowledged this mistake, and corrected it. It can be found corrected on lines 69, 81, 97, 145, and 295.
Line 108-109: Please include the conditions of freeze-drying (pressure and temperature)
The authors have acknowledged this mistake, and have added the required information on line 109.
Each figure should be mentioned in the text. For example, Figures 1 and 2 have no mentioned in the text.
The authors have acknowledged this mistake, and have added the required information on line 114 for the figure 1, on line 128 for figure 2, on line 249 for figure 3, and on line 312 for figure 4.
Line 134-136: Please use the symbol in the equations, not text
The authors have acknowledged this mistake, and have modified the equations accordingly. The changes can be found on lines 134-136.
In the Material and Method, section authors should include information about the statistical evaluation of data.
The authors had included this information previously on the subsection 2.6., which belongs to the materials and methods section, and therefore consider that this was already shown on the manuscript. On this subsection the statistical data analysis is explained when using CCD.
Why only Anova was performed for values in Tables 7 and 9. What about the data in Tables 1 and 4?
The authors performed an ANOVA test for the values on tables 1 and 4 but these are shown on tables 2 and 5, respectively.
Why the units of strength are expressed in g/cm2. The SI unit should be used as Pascal (N/m2)
The gel strength SI units are g/cm2. The authors would like to include a paper published on Marine Drugs where these units have been used Pretreatment Techniques and Green Extraction Technologies for Agar from Gracilaria lemaneiformis.
Line 182-185 and in the whole manuscript: Do not give the unit (%), just give the unit directly after the numbers.
The authors have modified the lines accordingly, and the changes can be found now on lines 190 to 192, and on lines 298 to 300.
Lines 505-507 in Conclusions: It is obvious without this study. Compare these two methods of extraction and give a suggestion of which method is better, and suggest the optimized condition of extraction as is written in the title of this paper.
The authors have acknowledged this correction, and have modified the conclusions accordingly on lines 531-532, and on lines 535-538.

Reviewer 2 Report
The article titled: :"Comparison study of an optimized ultrasound-based method versus an optimized conventional method for agar extraction, and protein co-extraction, from Gelidium sesquipedale" could be interest for Foods readers. A major revision is suggested.
Comments and recommendations of Reviewer are listed below:
Abstract:
- line 21 - it should be: "its" instead of: "their"
- line 29 - the conditions of extraction is not needed in abstract.
- lines 31-32 - which applications did the Authors mean - there is no information in the article (Conclusions).
Introduction:
- line 47 - why did the Authors use:"their". The sentence is about agar.
- lines 64-67 - the sentence is unclear (grammar, style) and should be rewritten.
- lines 69-70 - the style of the sentence should be corrected.
- lines 74-75 - the style of the sentence should be corrected.
- lines 80-82 - the style of the sentence should be corrected.
Materials and methods:
- point 2.3 line 107 - please explain the procedure of gel squeezing - it is not possible to obtain samples with the same water content - the water content should be checked and presented.
- point 2.4 line 125 - Table 2 does not contain data of extraction time and ultrasound powers - please correct (Do the Authors mean: Table 4?)
- p.2.7 - please give more details about the methodology.
Results and discussion:
- lines 241-242 - the sentence is incorrect and should be rewritten.
- line 292 - Do the Authors mean Table 2? (tables should be rearranged with changed an correct numeration).
- line 300 - it should be: "ultrasound treatment".
- lines 370-374 - please specify the solvent that was used.
- line 376 - please explain, why reextraction was negatively affected by the increase of temperature.
- lines 394-396 - please explain this phenomenon.
- p. 3.2.2.3. -it should be rather in: "Conclusions" chapter.
- p. 3.3. - please change the title of the chapter to more detailed.
- lines 415-416 - the sentence is incorrect (style) - please rewrite it.
- Please explain what the reason was for collecting FT-IR spectra. Did the Authors expect some differences in the spectral shape? Which differences?
- line 418 - compounds can not be identified based on FT-IR spectra, but characteristic bonds - please rewrite the sentence.
- Please give additional information about the region of 3000-3600 cm-1 in FT-IR spectra.
- lines 442-443 - what can be a reason for such situation?
- line 483 - the samples should be numbered with specific abbreviations. Please use abbreviation for certain samples along the manuscript.
- Please explain, why samples treated with ultrasounds are expected to be richer in sulphate groups.
- Table 7 - please chech carefully the statistic results (were the homogenous groups defined properly?).
- line 477-478 - the style is incorrect, please rewrite the sentence.
- Discussion must be more detailed. More results of other Authors should be presented.
Conclusions:
- It has been written (line 383), that extraction power did not affect significantly the extraction yield. In conclusions (line 506) it is written, that power of US can result in higher yield of agar extraction - please explain.
- Based on the obtained results in the study, please define, which conditions are the best for certain substances extraction.
Author Response
The authors appreciate the comments made by Reviewer 2, and wish to acknowledge each of them in the following points. On bold letters the corrections from the reviewer can be found, and following each statement the answer to those corrections. All the corrections have been included using the track mode.
Reviewer 2:
The article titled :"Comparison study of an optimized ultrasound-based method versus an optimized conventional method for agar extraction, and protein co-extraction, from Gelidium sesquipedale" could be interest for Foods readers. A major revision is suggested.
Comments and recommendations of Reviewer are listed below:
Abstract:
- line 21 - it should be: "its" instead of: "their"
The authors have acknowledged this correction, and it can be found now on line 21.
line 29 - the conditions of extraction is not needed in abstract.
The authors wish to inform the reviewer that these conditions are the ones corresponding to the optimized extraction processes, and therefore these are relevant to be included in the abstract; since these are ones of the main results of this paper.
- lines 31-32 - which applications did the Authors mean - there is no information in the article (Conclusions).
The author have acknowledged this correction, and have included line 531-532 including the requested information.
Introduction:
- line 47 - why did the Authors use: "their". The sentence is about agar.
The authors have acknowledged this correction, and it can be found now on line 47.
lines 64-67 - the sentence is unclear (grammar, style) and should be rewritten.
The authors have acknowledged this correction, and it can be found now on lines 64-67.
lines 69-70 - the style of the sentence should be corrected.
The authors have acknowledged this correction, and it can be found now on line 69-70.
lines 74-75 - the style of the sentence should be corrected.
The authors have acknowledged this correction, and it can be found now on lines 74-75.
lines 80-82 - the style of the sentence should be corrected.
The authors have acknowledged this correction, and it can be found now on line 80-82.
Materials and methods:
- point 2.3 line 107 - please explain the procedure of gel squeezing - it is not possible to obtain samples with the same water content - the water content should be checked and presented.
The authors have acknowledged this correction, and have added more information accordingly on line 107. Nevertheless, the authors wish to inform the reviewer that the extracts were completely dried after the squeezing step by freeze-drying, and therefore variations on the water content based on the squeezing step are not considered as relevant.
- point 2.4 line 125 - Table 2 does not contain data of extraction time and ultrasound powers - please correct (Do the Authors mean: Table 4?)
The authors have acknowledged this correction, and have modified line 125 accordingly.
- p.2.7 - please give more details about the methodology.
The authors have acknowledged this correction, and have added information on lines 161-166.
Results and discussion:
- lines 241-242 - the sentence is incorrect and should be rewritten.
The authors have acknowledged this correction, and have changed line 247 accordingly.
- line 292 - Do the Authors mean Table 2? (tables should be rearranged with changed an correct numeration).
The authors meant table 4, as table 4 shows the conditions of UAE as cited on line 296.
- line 300 - it should be: "ultrasound treatment".
The authors have acknowledged this correction, and have changed line 304 accordingly.
- lines 370-374 - please specify the solvent that was used.
The solvent used for protein and agar extraction is specified on the materials and methods sections 2.3 and 2.4, which is distilled water.
- line 376 - please explain, why reextraction was negatively affected by the increase of temperature.
The authors have acknowledged this correction, and the changes can be seen now on line 384 stating that the temperature had no significant effect.
lines 394-396 - please explain this phenomenon.
The authors have added information on line 404 according to the reviewer’s suggestion.
- p. 3.2.2.3. -it should be rather in: "Conclusions" chapter.
The authors have acknowledged this correction, and have added the information previously shown on 3.2.2.3 now on the conclusions.
p. 3.3. - please change the title of the chapter to more detailed.
The authors have acknowledged this correction, and it can be found now on line 421.
- lines 415-416 - the sentence is incorrect (style) - please rewrite it.
The authors have acknowledged this correction, and have modified line 415 accordingly.
- Please explain what the reason was for collecting FT-IR spectra. Did the Authors expect some differences in the spectral shape? Which differences?
The authors have acknowledged this correction, and have added information on lines 423-425.
- line 418 - compounds can not be identified based on FT-IR spectra, but characteristic bonds - please rewrite the sentence.
The authors have acknowledged this correction, and have corrected line 430 accordingly.
- Please give additional information about the region of 3000-3600 cm-1 in FT-IR spectra.
The authors have acknowledged this correction, and have added information on lines 436-438.
- lines 442-443 - what can be a reason for such situation?
The authors hypothesized that shorter treatment times would lead to less agar degradation, but ultrasound may have a negative effect on the agar structure for gelling purposes. This will be investigated in future works. Besides, further explanations were given on the following lines in the manuscript.
- line 483 - the samples should be numbered with specific abbreviations. Please use abbreviation for certain samples along the manuscript.
The authors have acknowledged this correction, and have modified the information on table 7.
- Please explain, why samples treated with ultrasounds are expected to be richer in sulphate groups.
The authors have acknowledged this correction, and have modified the information on lines 463-465.
Table 7 - please chech carefully the statistic results (were the homogenous groups defined properly?).
The statistical results provided have been crosschecked, and no mistakes have been found.
line 477-478 - the style is incorrect, please rewrite the sentence.
The authors have acknowledged this correction, and have modified line 494-495 accordingly.
- Discussion must be more detailed. More results of other Authors should be presented.
The authors would like to inform the reviewer that the novelty of the study makes it complex to be compared to other studies, as there is a very limited amount of papers that could be of benefit for the discussion.
Conclusions:
- It has been written (line 383), that extraction power did not affect significantly the extraction yield. In conclusions (line 506) it is written, that power of US can result in higher yield of agar extraction - please explain.
The authors have acknowledged this correction, and have modified line 523-524 accordingly.
- Based on the obtained results in the study, please define, which conditions are the best for certain substances extraction.
The authors have acknowledged this correction, and have included this on line 524.

Reviewer 3 Report
Very well designed study and clearly presented data. As you mentioned in your discussion the de-polymerization effect of US should be more deeply studied in the future experiments.
Author Response
Reviewer 3:
Very well designed study and clearly presented data. As you mentioned in your discussion the de-polymerization effect of US should be more deeply studied in the future experiments.
The authors appreciate the positive feedback given by reviewer 3, and would like to thank reviewer 3 for it. A new experiment to study this de-polimerization effect has been planned, and will be carried out by the authors.

Round 2
Reviewer 1 Report
The authors corrected the manuscript accordingly.
Reviewer 2 Report
I would like to thank the Authors for taking into account comments and recommendations of the Reviewer.
Please explain, what do the Authors mean by the statement, that: "the extracts were completely dried" - (the answer for point 2.3, line 107)